# Mitigating Effect of *Lindera obtusiloba* Blume Extract on Neuroinflammation in Microglial Cells and Scopolamine-Induced Amnesia in Mice

**DOI:** 10.3390/molecules26102870

**Published:** 2021-05-12

**Authors:** Song-Hee Jo, Tae-Bong Kang, Sushruta Koppula, Duk-Yeon Cho, Joon-Soo Kim, In-Su Kim, Dong-Kug Choi

**Affiliations:** 1Department of Applied Life Science & Integrated Bioscience, Graduate School, BK21 Program, Konkuk University, Chungju 27478, Korea; wowsong333@naver.com (S.-H.J.); kangtbko@kku.ac.kr (T.-B.K.); ejrdus1026@naver.com (D.-Y.C.); kgfdkr@gmail.com (J.-S.K.); 2Department of Integrated Bioscience & Biotechnology, College of Biomedical and Health Science, Research Institute of Inflammatory Disease (RID), Konkuk University, Chungju 27478, Korea; 3Department of Biotechnology, College of Biomedical & Health Science, Konkuk University, Chungju 27478, Korea; sushrutak@gmail.com

**Keywords:** *Lindera obtusiloba*, neuroinflammation, BV2 cell

## Abstract

*Lindera obtusiloba* Blume (family, Lauraceae), native to Northeast Asia, has been used traditionally in the treatment of trauma and neuralgia. In this study, we investigated the neuroinflammatory effect of methanol extract of *L. obtusiloba* stem (LOS-ME) in a scopolamine-induced amnesia model and lipopolysaccharide (LPS)-stimulated BV2 microglia cells. LOS-ME downregulated the expression of inducible nitric oxide synthase (iNOS), cyclooxygenase (COX)-2, inflammatory cytokines, and inhibited the phosphorylation of nuclear factor kappa-B (NF-ĸB) and extracellular signal-regulated kinase (ERK) in LPS-stimulated BV2 cells. Male C57/BL6 mice were orally administered 20 and 200 mg/kg of LOS-ME for one week, and 2 mg/kg of scopolamine was administered intraperitoneally on the 8th day. In vivo behavioral experiments (Y-maze and Morris water maze test) confirmed that LOS-ME alleviated cognitive impairments induced by scopolamine and the amount of iNOS expression decreased in the hippocampus of the mouse brain. Microglial hyper-activation was also reduced by LOS-ME pretreatment. These findings suggest that LOS-ME might have potential in the treatment for cognitive improvement by regulating neuroinflammation.

## 1. Introduction

Alzheimer’s disease (AD) is a progressive neurodegenerative disease with a wide range of clinical complications associated with memory loss and impairment of functional neurons leading to neuronal cell death [1,2]. Most cases of AD are late-onset and sporadic, with no proven evidence for a Mendelian pattern of inheritance. The prevalence rate of the disease increases with life expectancy and affects more than one-third of people over the age of 70 [1]. AD is characterized by impairments in hippocampal neurogenesis, associated cognitive dysfunction, and memory deficits [2]. The normal cholinergic system in the brain influences hippocampal neurogenesis and cognitive function by modulating neurogenic mechanisms such as those involving brain-derived neurotrophic factor (BDNF) and cAMP response element-binding protein (CREB) [3]. Subsequently, newly-formed neural progenitor cells at the dentate gyrus of the hippocampus grow as new functional neurons and integrate with the existing neuronal circuit, thereby act as crucial in cognition and memory [2].

Microglia are a type of glial cell located throughout the brain and account for about 10–20% of all cells found within the brain [4]. These resident immune cells, which act as the first form of immune defense systems in the central nervous system (CNS), remain steady by interacting with the cell surface and soluble factors from neighboring cells [5]. However, neuroinflammation is initiated by microglia upon activation following exposure to pathogen-associated molecular patterns (PAMPs) and/or endogenous damage-associated molecular patterns (DAMPs), including the endotoxin lipopolysaccharide (LPS) released from certain bacteria. Activated microglia exhibits diverse effects on the progression of AD. Preclinical and clinical studies indicated that activated microglia triggers the release of various proinflammatory factors, such as reactive oxygen species (ROS), interleukin-6 (IL-6), IL-1β, nitric oxide (NO), and other inflammatory cytokines [6]. Microglial activation plays a broad role as a causative factor for a range of neurological disorders [7]. Thus, microglia occupy a central position in the defense and maintenance of the CNS, attracting interest throughout the scientific field as a potential therapeutic target in controlling or preventing neurological disorders and recovery from brain injury.

*Lindera obtusiloba* Blume (*L. obtusiloba*) from the family Lauraceae is a native Northeast Asian deciduous shrub traditionally used in the treatment of trauma, neuralgia, liver disorders, sleeplessness, anxiety, and cardiovascular defects [8]. Ethnopharmacological reports revealed that *L. obtusiloba* has been used in the treatment of allergic diseases, including rhinitis, asthma, and atopic dermatitis [9]. In addition, studies also indicated that *L. obtusiloba* has antidepressant effects [10], antifibrotic [11], neuroprotective [12], anti-inflammatory [13], and antihepatotoxic effects [14]. However, works on the efficacy and scientific evaluation of *L. obtusiloba* on microglia-mediated neuroinflammation and cognitive impairments have not been studied.

It is well known that the anti-cholinergic drug scopolamine can cause learning and memory deficits by disrupting cholinergic neurotransmission and has been widely used to induce amnesia in experimental rodent models for screening of memory-enhancing agents [15]. Therefore, in this study, the effects of *L**. obtusiloba* stem extract against LPS-induced microglia-mediated neuroinflammation in vitro and scopolamine-induced amnesic mice model in vivo were investigated by using various parameters.

## 2. Results

### 2.1. Effect of LOS-ME on NO Production in LPS-Activated BV2 Cells

To determine whether the decrease in NO production in BV2 microglia cells is due to the toxicity of BV2 microglia cells by LOS-ME, 3-(4,5-dimethylthiazol-2-yl)-2,5-diphenyltetrazolium bromide (MTT) assay was performed. As a result, there was no significant effect on the viability of BV2 cells in all groups (Figure 1A). To investigate the effect of LOS-ME on LPS-induced NO production in BV2 microglia cells, LOS-ME (25, 100, 200 μg /mL) was treated with LPS (200 ng/mL) on BV2 microglia cells. After stimulation for 24 h, the culture medium supernatant was confirmed by reacting with the Griess reagent. The amount of NO released from the control group was 2.7 μM and the LPS group increased about nine-fold to 23.5 μM, and the 25, 100, 200 μg/mL group treated with LPS reduced NO emission by 29.2%, 64.7%, and 74.8%, respectively (Figure 1B). The IC_50_ value is 92.4 ± 0.1 μg/mL. Therefore, LOS-ME concentrations of 25, 100, and 200 μg/mL were used in the following experiments.

### 2.2. Effect of LOS-ME on iNOS, COX-2 and Proinflammatory Cytokine Expression in LPS- Activated BV2 Cells

We investigated the effect of LOS-ME on inhibition of iNOS, COX-2, and proinflammatory cytokine (TNF-α, IL-1β, IL-6) expression at the mRNA level and protein level in LPS-induced BV2 microglia. LOS-ME at the indicated concentrations (25, 100, 200 μg/mL) with or without LPS (200 ng/mL) were treated in BV2 cells. Expression of iNOS, COX2, and proinflammatory cytokines (TNF-α, IL-1β, IL-6) at the mRNA level was determined using RT-PCR analysis obtained by stimulating LPS for 6 h (Figure 2A,C and Figure 3A–C). The expression of iNOS and COX-2 at the protein level was confirmed by Western blot of BV2 cells obtained by stimulating LPS for 18 h (Figure 2B,D). In the previous experiment, it was confirmed that LOS-ME inhibited NO production in a concentration-dependent manner (Figure 1A), and it was confirmed that iNOS expression at the mRNA level was decreased in a concentration-dependent manner (Figure 2A). In addition, it was confirmed that the expression of iNOS at the protein level was also decreased depending on the LOS-ME concentration (Figure 2B).

COX-2 is the predominant enzyme at sites of inflammation [16]. Studies have shown that the product PGE2 by COX-2 is a common mediator of brain inflammation. It has also been shown that PGE2 is released after stimulation of microglia with LPS [17]. Therefore, we confirmed the effect of LOS-ME on COX-2 expression in LPS-activated BV2 microglia cells. As a result, the expression of COX-2 at the mRNA level was decreased in a LOS-ME concentration-dependent manner (Figure 2C). In addition, it was confirmed that the expression of COX-2 was decreased depending on the LOS-ME concentration at the protein level (Figure 2D). We also confirmed the effect of LOS-ME on the expression of TNF-α, IL-1β, and IL-6, proinflammatory cytokines that play a central role in various inflammations in LPS-stimulated BV2 microglia. As a result of the experiment, it was confirmed that TNF-α, IL-1β, and IL-6 production decreased significantly with increasing concentration of LOS-ME (Figure 3A–C).

### 2.3. Effect of LOS-ME on NF-kB Pathway Phosphorylation in LPS-Activated BV2 Cells

LPS stimulation of TLR4 induces transcription of proinflammatory cytokines by activating the NF-κB pathway, which plays an important role in the activation of BV2 microglia [18]. Therefore, it was confirmed that LOS-ME inhibited the phosphorylation of the p65 subunit of NF-κB in BV2 microglia cells. BV2 microglia cells were treated with LOS-ME for 30 min with LPS. As a result, it was confirmed that the phosphorylation of p65 decreased compared to the LPS-only treatment group according to the concentration of LOS-ME (Figure 4A). In addition, to confirm the translocation of p65 to the nucleus, immunocytochemistry (ICC) analysis was performed. Compared to the LPS group, it was confirmed that the nuclear migration of p65 decreased when LOS-ME 200 μg/mL was treated with LPS (Figure 4B).

### 2.4. Effect of LOS-ME on MAPKs Phosphorylation in LPS-activated BV2 CELLS

MAPKs signaling is known to play an important role in the regulation of proinflammatory cytokine production in activated BV2 cells [19,20]. In order to determine whether LOS-ME is involved in the phosphorylation of ERK, JNK, and p38 in MAPKs signaling in LPS-stimulated BV2 microglia cells, LPS and LOS-ME were treated together for 30 min and examined through Western blot. As a result, phosphorylation of ERK was inhibited by LOS-ME in a concentration-dependent manner (Figure 5A). However, LOS-ME was not involved in JNK phosphorylation, and p38 phosphorylation was slightly decreased when compared with the LPS group (Figure 5B).

### 2.5. Effect of LOS-ME on Cognitive and Behavioral Impairments in Scopolamine-Induced Mouse Model

To investigate the effects of LOS-ME on cognition and behavioral impairments, Y-maze and Morris water maze tasks were conducted with 20 mg/kg of LOS-ME administered orally for 7 days. As a positive control, tacrine (10 mg/kg, p. o.) was administered using the same treatment protocol (Section 4.4). Y-maze behavior test showed similar activity levels in all groups (Figure 6A). As a result of calculating the non-overlapping number by measuring the visited branch by numbering each branch, it was confirmed that the memory of the LOS-ME group was improved compared to the scopolamine group (Figure 6B). In the Morris water maze behavior experiment, it was confirmed that the LOS-ME treated mice escaped to the platform faster than the scopolamine group (Figure 6C). Based on these results, it was confirmed that LOS-ME alleviated cognitive and behavioral disorders in scopolamine-induced amnesia. The positive control group treated with tacrine also exhibited similar effects at 10 mg/kg dose.

### 2.6. Effect of LOS-ME on iNOS Expression in Hippocampus of Mice Brain Tissue

LOS-ME (20 mg/kg) was administered orally for 7 days, and scopolamine was administered intraperitoneally after the final LOS-ME pretreatment. Then only the hippocampus region of the brain was isolated, and iNOS and COX-2 expression were measured through Western blot. As a result, it was confirmed that the expression of iNOS decreased in the LOS-ME (20 mg/kg) treated group significantly (Figure 7A). However, LOS-ME (20 mg/kg) did not affect the expression of COX-2 in the hippocampus (Figure 7B).

## 3. Discussion

It is well documented that the inflammatory mediators in the CNS causing neuroinflammation is one of the major pathogenic factors in neurodegenerative diseases such as Parkinson’s disease (PD), Alzheimer’s disease (AD), Huntington’s disease (HD), and amyotrophic lateral sclerosis (ALS) [21,22,23,24]. Neuroinflammation by microglial activation is a major event seen in neurodegenerative disorders and indeed affects the overall behavioral and cognitive functions [25]. Therefore, agents that alleviate the inflammatory processes within the brain might help in mitigating these disease conditions or delay the progression of the disease. In this study, LOS-ME significantly inhibited LPS-activated BV2 microglia-mediated neuroinflammatory responses in the in vitro experiments and mitigated the scopolamine-induced amnesia by ameliorating the behavioral and cognitive impairments in mice.

In preliminary studies, we investigated whether various concentrations of LOS-ME treated with LPS to BV2 cells can affect the nitric oxide (NO) production inhibition rate and cell survival rate. Data revealed that LPS-induced NO production was reduced significantly and dose-dependently in LOS-ME-treated BV2cells, and there was no effect on cell viability. Further, it was confirmed that LOS-ME significantly reduced the LPS-stimulated increased iNOS expression in a concentration-dependent manner at both mRNA and protein levels. COX-2 a dominant enzyme in the site of inflammation, and its product PGE2 are common mediators of brain inflammation. It has also been shown that PGE2 is released after stimulation of microglia with LPS [26]. In this study, LOS-ME showed attenuating effects on the expression of COX 2 at the mRNA and protein level in LPS-activated BV2 cells.

We also confirmed the effect of LOS-ME on the expression of proinflammatory cytokines such as TNF-α, IL-1β, and IL-6, which play a central role in various inflammatory processes in LPS-stimulated BV2 cells. Our results confirmed that the increased production of TNF-α, IL-1β, and IL-6 in LPS-stimulated BV2 cells was significantly decreased by LOS-ME in a concentration-dependent fashion.

LPS-induced activation of TLR4 induces transcription of proinflammatory cytokines by activating the NF-κB pathway, which plays an important role in the activation of BV2 microglia [18]. To understand the role of LOS-ME in the NF-κB pathway, the translocation and phosphorylation of the NF-κB p65 subunit were measured in LPS-stimulated BV2 cells. IHC data revealed that LOS-ME (200 µg/mL) inhibited the phosphorylation of p65 and also reduced the nuclear migration at high concentrations. Our results are in agreement with earlier reported works that L. obtusiloba inhibited the phorbol 12-myristate 13-acetate and calcium ionophore A23187-stimulated gene expression of proinflammatory cytokines (TNF-α and IL-6) in human mast cells, and the anti-inflammatory effects were NF-κB dependent [9].

MAPKs signaling is known to play an important role in the regulation of proinflammatory cytokine production in activated BV2 cells [19,20]. Therefore, we evaluated whether LOS-ME is involved in the phosphorylation of ERK, JNK, and p38. Although LOS-ME did not influence the phosphorylation of JNK and p38 phosphorylation, the ERK was inhibited in a concentration-dependent manner (Figure 5A). Why the effect of LOS-ME was observed only in regulating ERK phosphorylation and not in other MAPKS signaling needs further study. In the present study, LOS-ME significantly downregulated the LPS-induced production of proinflammatory cytokines and further regulated the ERK MAPKs signaling in BV-2 microglial cells indicating its potentially beneficial role in suppressing neuroinflammation. In vivo behavioral experiments such as Y-maze and Morris water maze were conducted to confirm whether LOS-ME has an effect on cognitive impairment in the scopolamine-induced amnesic mice model. Previous studies revealed that scopolamine is commonly employed to induce cognitive and memory impairment in experimental models of neurodegenerative disorders [27]. Further, increased expression of inflammatory cytokines and glial hyperactivity was observed along with memory impairment in scopolamine-induced animal models [28]. Y-maze behavior test showed similar activity levels in the total number of arm entries in all groups. However, the percentage of spontaneous alterations was significantly improved in the scopolamine-induced LOS-ME treated group compared with the scopolamine alone treated group indicating improved memory and behavioral pattern. Morris water maze behavior experiment also confirmed that the scopolamine-induced LOS-ME group escaped to the platform faster than the scopolamine alone treated group indicating an improved special memory.

An earlier report indicated that *L. obtusiloba* showed neuroprotective properties against glutamate-induced toxicity in HT22 hippocampal cells [12]. Based on the in vivo data and earlier reports, we believe that LOS-ME is effective in improving cognition with neuroprotective effects. Since tacrine has been shown to improve cognitive functions in experimental animal models of AD [29,30]. We used tacrine as a positive reference drug in our study. In agreement, tacrine treatment (10 mg/kg) to scopolamine-induced groups also exhibited protective effects that were similar to LOS-ME treated at 20 mg/kg dose.

Earlier reports indicated the use of *L. obtusiloba* at the dose range of 50–2000 mg/kg bodyweight for 7–21 days in various in vivo animal models, including the effect of *L. obtusiloba* on cancer metastasis [31], improving physical performance in diabetic db/db mice [32], antidepressant effects [10], hepatoprotective effects [14] and in the treatment of in allergic asthma [33]. In our present study, the dose of LOS-ME used (20 mg/kg) was markedly low in comparison with the published reports in ameliorating the scopolamine-induced amnesia in mice. Further, the overall well-being of the experimental animals was well maintained and tolerated with no untoward side effects during the LOS-ME treatment and course of the study.

It is well documented that scopolamine administration influences the excessive release of proinflammatory mediators. The expression of iNOS and COX2 was analyzed in the hippocampus, which plays a role in brain learning, memory, and recognition of a new environment. Data showed that the increased iNOS expression in scopolamine-induced mouse hippocampus was reduced by LOS-ME (20 mg/kg). Further, scopolamine or LOS-ME and the positive control tacrine did not show any effect on the overall expression of COX2. However, in-depth studies using various dose ranges and the treatment duration of LOS-ME using various cognitive experimental models are warranted to explore the significance and dose-dependent effects of LOS-ME.

Previous reports on the phytochemicals present in *L. obtusiloba* indicated an array of compounds, including flavonoids, butenolides, lignans, and neolignans [34]. Further, the presence of active principles such as quercitrin, afzelin, (+)-syringaresinol, linderin A, (+)-episesamin, secoisolariciresinol derivatives were also well documented [35]. These compounds were well reported for their potential pharmacological effects, including antioxidant, antiatherosclerotic, antihistamine, antiplatelet, anti-inflammatory, and neuroprotective effects [36]. The active constituents present in the LOS-ME extract might act individually or in a synergistic way in exhibiting the potential ameliorating effects in microglia-mediated neuroinflammation and antiamnesic effects. However, further studies on the insolation of active principles present in LOS-ME and their underlying mechanisms are necessary.

In conclusion, based on in vitro and in vivo data, our present study provided scientific support that LOS-ME might be used as a potential therapeutic remedy in the treatment of neuroinflammation-mediated disorders and cognition dysfunctions seen in neurodegenerative diseases.

## 4. Materials and Methods

### 4.1. Reagents

Lipopolysaccharide (LPS), dimethyl sulfoxide (DMSO), 3-(4,5-dimethyl-2-thiazolyl)-2,5-diphenyl-2H-tetrazolium bromide (MTT), sulfanilamide, *N*-(1-naphthyl) ethylenediamine dihydrochloride, scopolamine hydrobromide, 9-Amino-1,2,3,4-tetrahydroacridine hydrochloride hydrate (tacrine), chloroform, isopropanol, and 2-mercaptoethanol were purchased from Sigma-Aldrich (St. Louis, MO, USA). 10X RIPA buffer was acquired from Millipore (Burlington, MA, USA). Protease and phosphatase inhibitor cocktail tablets were obtained from APEx Bio (Boston, MA, USA). Dulbecco’s modified Eagle’s medium (DMEM) and phosphate-buffered saline (PBS) were obtained from Welgene (Gyeongsang-do, South Korea). Fetal bovine serum (FBS), penicillin/streptomycin (P/S), and 0.5% trypsin-EDTA (TE) were obtained from Gibco-BRL Technologies (Carlsbad, CA, USA). Glycine, sodium chloride, ethanol 95.0%, and methyl alcohol 99.5% were purchased from Samchun (Seoul, South Korea). Phosphoric acid was acquired from Duksan (Gyeonggi-do, South Korea). Trizol was obtained from Ambion (Europe). Agarose EP Master, 50X TAE, DEPC-treated water, 20% SDS solution were obtained Biosesang (Gyeonggi-do, South Korea).

### 4.2. Preparation of Methanol Extract of L. obtusiloba Stem

*L. obtusiloba* stem (LOS) was collected in Hanyang, South Korea, in December 2019, and the authenticity of the material was confirmed by a taxonomist at Konkuk University, South Korea, and stored in our department herbarium. For extraction, dried *L. obtusiloba* stem approximately 100 g was mixed with 1 L of 99.5% methanol and kept in a shaking incubator at 25 °C for three days. The precipitate was removed by filtering using filter paper (Whatman, Maidstone, UK), and the solution was under vacuum pressure using a rotary evaporator to remove the entire methanol at 45 °C. The final extract named LOS-ME hereafter was stored in a −80 °C freezer for 24 h and freeze-dried until the experiments.

### 4.3. Cell Culture, Cell Viability, and NO Assay

BV2 microglial cells were cultured in an incubator maintained at 37 °C, 5% CO_2_ using DMEM containing 5% FBS and 1% penicillin/streptomycin. For cell viability and NO assay, BV2 cells were seeded in a 24 well plate at a density of 3.0 × 10⁵ cells/mL, treated with various concentrations of LOS-ME (25, 100, 200 μg/mL) and LPS (200 ng/mL), and incubated. After 18 h, 100 μL of the supernatant and 100 μL of the Griess reagent (1% sulfanilamide and 0.1% naphthylamide in 5% phosphoric acid) were mixed and reacted, and absorbance was measured at 540 nm using SUNRISE (Tecan GmbH, Männedorf, Switzerland). The amount of NO produced is based on a standard curve measured using sodium nitrite (145 μM, Sigma-Aldrich, Saint Louis, MO, USA). For viability test, 20 μL of MTT solution (2 mg/mL, methylthiazolyldiphenyl-tetrazolium bromide in PBS) was added into the remaining 400 μL of the previously seeded 24-well plate, wrap with foil, and incubate. After 1 h, the supernatant was suctioned, and 400 μL of DMSO was treated to dissolve formed formazan crystals, and then 200 μL of each was transferred to a 96-well plate. The absorbance was measured at 552 nm using SUNRISE.

### 4.4. Animals and Treatment

Male C57BL/6 mice (8 weeks, 20–25 g) were supplied from Deahan Bio Link (Chungcheong-do, South Korea). All experiments were performed in accordance with the Principles of Laboratory Animal Care (NIH publication no.85-23, revised 1985) approved animal protocols and guidelines established by the Konkuk University, IACUC (no. KU20210, 13/01/2021). The animals were housed in a controlled environment maintain 23 ± 1 °C temperature and 50% ± 5% humidity, and food and water were supplied ad libitum. The room lights were turned on 12 h between 8:00 and 20:00 h. After animals were acclimatized to laboratory conditions for 2 weeks, they were randomly divided into experimental groups. The animal experimental study design is presented in Figure 6A. In brief, the animals were divided into four groups: Control group (*n* = 8; 0.9% saline, i.p.), scopolamine 2 mg/kg, (*n* = 8; scopolamine in 0.9% saline, i.p.), LOS-ME (*n* = 8; LOS-ME 20 mg/kg, p.o. + scopolamine 2 mg/kg i.p.) and tacrine (*n* = 8; tacrine 10 mg/kg, p.o. + scopolamine 2 mg/kg i.p.). LOS-ME and tacrine were administered using an oral gavage dissolved in 0.9% saline containing 1% Tween 80 and 0.9% saline, respectively, for 7 days prior to scopolamine injection.

#### 4.4.1. Y-Maze Test

The Y-maze test was used to determine the spontaneous movement test and the cognitive memory test. The test used a Y-shaped maze with three identical arms separated at an angle of 120°, and each arm had a different environment and numbered. The mice were placed in the center of the maze and allowed to roam freely for 1 min, observed and acclimated, then measured for 3–5 min. The number of arms visited was recorded in order.

#### 4.4.2. Morris Water Maze Test

The Morris water maze test is used to measure spatial learning and cognitive memory tests. The test was placed in a circular pool (diameter 90 cm) filled with water containing white organic paint, with the platform submerged and invisible in one of the pool quadrants. The temperature of the water was maintained at 20–22 °C. For three days, the mouse was adapted to find the escape platform and climb up. In the measurement, the time to climb on the platform for 2 min was recorded using the video camera-based SMART3.0 software (Panlab, Barcelona, Spain).

### 4.5. Total RNA Isolation and Reverse Transcription Polymerase Chain Reaction (RT-PCR)

BV2 cells were seeded in a 6-well plate at a density of 2.0 × 10^5^ cells/mL and treated with LOS-ME 25, 100, 200 μg/mL, and LPS 200 ng/mL. The total RNA from the treated cells was acquired using Trizol reagent (Invitrogen Life Technologies, Waltham, MA, USA) according to the manufacturer’s instructions. The cDNA synthesis was performed by measuring the concentration of total RNA using a NanoDrop machine (Molecular Devices, San Jose, CA, USA), followed by reverse transcription using reverse transcription reagent (Promega, Madison, WI, USA) and PCR equipment (Bio-Rad, San Francisco, CA, USA) according to the manufacturer’s instructions. Then, using specific primers, cDNA was amplified by PCR following the method described previously [2]. Electrophoresis was performed with a 1–2% agarose gel using Mupid-2plus (TaKaRa, Kyoto, Japan) bands were photographed with a Gel Imaging System (Davinch-K, Seoul, Korea) and then calibrated through ImageJ software.

### 4.6. Western Blot

BV2 cells were seeded in a 6-well plate at a density of 3.0 × 10^5^ cells/mL and treated with LOS-ME 25, 100, 200 μg/mL, and LPS 200 ng/mL for 18 h. The cells were washed with PBS twice and lysed using 1X RIPA lysis buffer. Cell lysates were centrifuged at 14,000 rpm, and the supernatants were collected for analysis. The protein concentration of each sample was obtained and normalized using a DC Protein Assay kit (Bio-Rad, San Francisco, CA, USA). Equal amounts of proteins (20–40 μg) were loaded and separated electrophoretically in 10% sodium dodecyl sulfate-polyacrylamide gel and were transferred to polyvinylidenedifuoride membranes (Millipore, Bedford, MA, USA). After blocking the transferred membrane with BSA for 1 h, the membranes were washed and incubated overnight at 4 °C with primary antibodies to anti-iNOS (1:2000) (BD Biosciences, CA, USA), anti-COX2 (1:2000), anti-phospho-CREB (1:1000) (Abcam, Cambridge, UK), anti-β-actin (1:10000) (Sigma-Aldrich, Saint Louis, MO, USA), anti-JNK, anti-phospho-JNK, anti-ERK, anti-phospho-ERK, anti-p38, anti-phospho-p38, anti-p65, anti-phospho-p65, anti-CERB, anti-IĸBα, anti-phospho- IĸBα (1:1000) (Cell signaling), anti-BDNF (1:1000) (NOVUS Biologicals, CO, USA). After incubation, the membranes were washed and incubated with secondary antibody (1:2000) (Bio-rad, Cell signaling) for 1 h at room temperature. The membrane was photographed using ImageQuant LAS500 (GE Healthcare, Chicago, IL, USA) and analyzed with ImageJ software.

### 4.7. Statistical Analyses

The data were expressed as mean ± standard deviation (SD) analyzed using a one-way ANOVA followed by Tukey’s multiple comparison test using GraphPad Prism 8.0.1 (San Diego, CA, USA) software. In all cases, probability values of *p* < 0.05 were considered statistically significant.

## Figures and Tables

**Figure 1 molecules-26-02870-f001:**
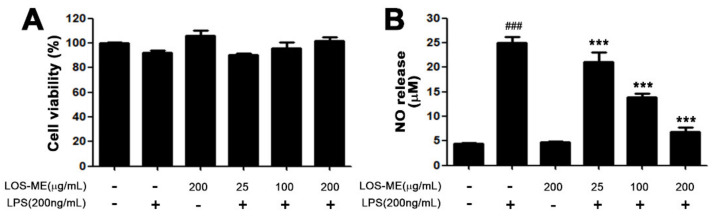
Effects of LOS-ME on cell viability and nitric oxide (NO) production in LPS-activated BV2 cells. BV2 cells were treated with LPS (200 μg/mL) and LOS-ME at different concentrations (25, 100, 200 μg/mL) and incubated for 24 h. (**A**) Cell viability was confirmed through MTT assay. (**B**) Supernatant was transferred and reacted with the Griess reagent to confirm the release of NO. Data are presented as mean ± SD (*n* = 6). ^###^ *p* < 0.001, compared with controls, *** *p* < 0.001, compared with LPS group by one-way ANOVA.

**Figure 2 molecules-26-02870-f002:**
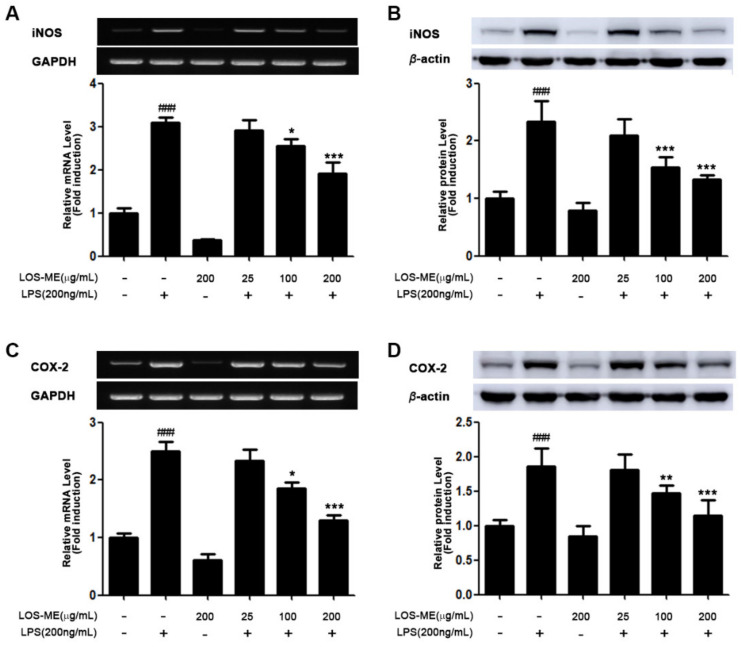
The effects of LOS-ME on iNOS and COX2 expression in LPS-activated BV2 microglial cells. (**A**) iNOS mRNA expression with corresponding fold change, (**B**) iNOS protein expression with corresponding fold change. (**C**) COX2 mRNA expression with corresponding fold change and (**D**) COX2 protein expression with corresponding fold change. GAPDH and β-actin were used as an internal control for mRNA and protein expression, respectively. Data are presented as mean ± SD (*n* = 3). ^###^
*p* < 0.001, compared with control * *p* < 0.05, ** *p* < 0.01, and *** *p* < 0.001 compare with LPS group by one-way ANOVA.

**Figure 3 molecules-26-02870-f003:**
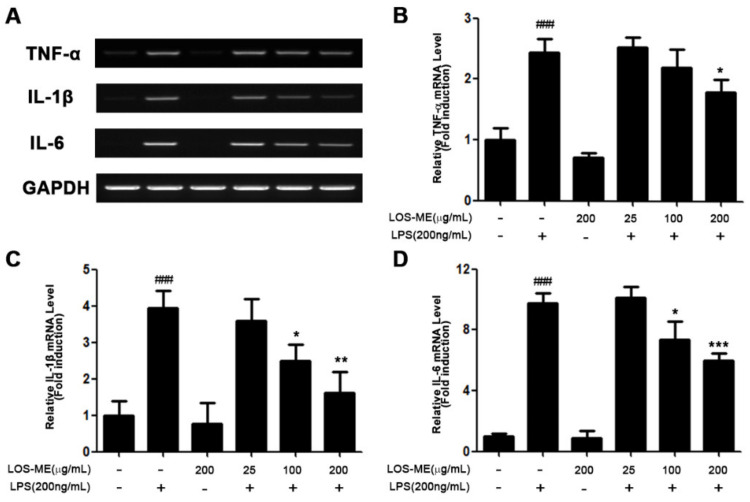
The effects of LOS-ME on TNF- α, IL-1β, and IL-6 mRNA expression in LPS-activated BV2 microglial cells. (**A**) Measurement of the expression level of proinflammatory cytokines (TNF-α, IL-1β, and IL-6) on mRNA level used RT-PCR (**B**)TNF-α mRNA expression with corresponding fold change and (**C**) IL-1β mRNA expression with corresponding fold change. (**D**) IL-6 mRNA expression with corresponding fold change. GAPDH was used as an internal control. Data are presented as mean ± SD (*n* = 3). ^###^ *p* < 0.001, compared with control and * *p* < 0.05, ** *p* < 0.01, and *** *p* < 0.001 compare with LPS group by one-way ANOVA.

**Figure 4 molecules-26-02870-f004:**
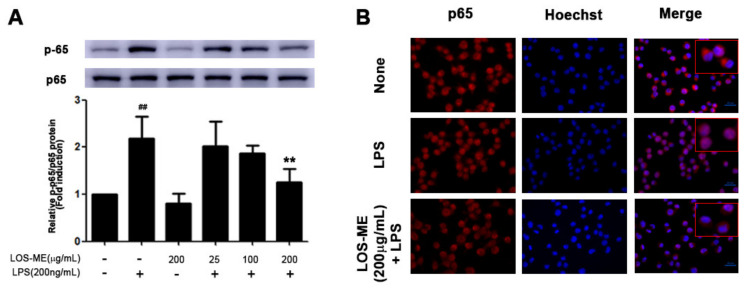
Effect of LOS-ME on NF-κB p65 phosphorylation in LPS-activated BV2 cells. BV2 microglia cells were treated with LOS-ME (200 μg/mL) and LPS (200 ng/mL) for 30 min and confirmed by Western blot. (**A**) protein expression of p65 and p-p65 with corresponding fold change. (**B**) BV2 microglia cells were treated with LOS-ME (200 µg/mL) and LPS (200 ng/mL) for 30 min and confirmed by ICC assay. Translocation of p65 protein was determined using an anti-p65 antibody and Alexa Fluor 568-labeled goat anti-mice antibody on immunofluorescence. Data are presented as mean ± SD (*n* = 3). ^##^ *p* < 0.01 compared with control group, ** *p* < 0.01 compared with LPS group by one-way ANOVA.

**Figure 5 molecules-26-02870-f005:**
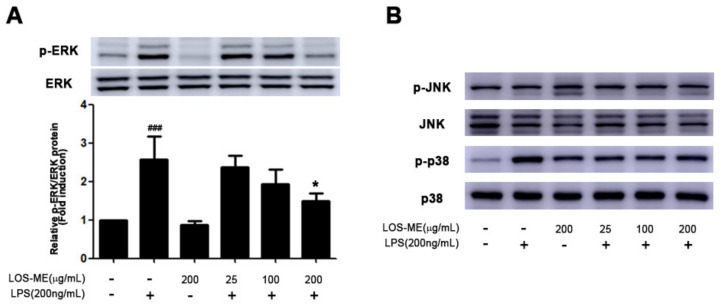
Effect of LOS-ME on ERK, JNK, and p38 phosphorylation in LPS-activated BV2 cells. BV2 microglia were treated with LOS-ME (25, 100, 200 μg/mL) and LPS (200 ng/mL) for 30 min and confirmed by attaching each phospho-form antibody. (**A**) Protein expression of ERK with corresponding fold change. (**B**) Protein expression of JNK and p38. The total form of each ERK, JNK, and p38 was used as an internal control. Data are presented as mean ± SD (*n* = 3). ^###^
*p* < 0.001 compared with control. * *p* < 0.05 and *** *p* < 0.001 compared with LPS group by one-way ANOVA.

**Figure 6 molecules-26-02870-f006:**
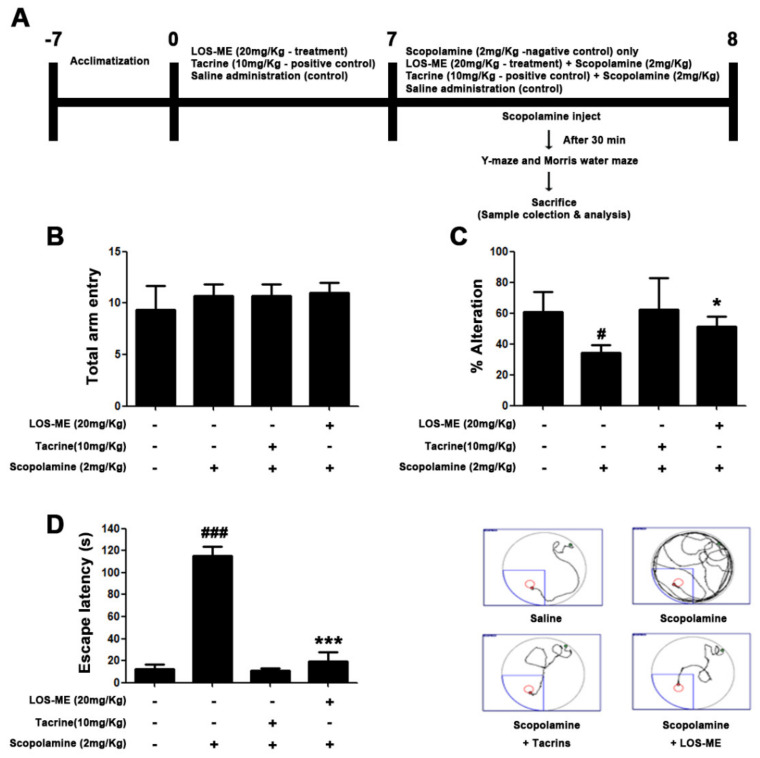
Effects of LOS-ME on cognitive and behavioral impairments in the scopolamine-induced amnesic mouse model. LOS-ME (20 mg/kg, p.o.) was dosed for 7 days, and on the 8th day, scopolamine was administered (2 mg/kg, i.p.). (**A**) Schematic representation of the animal experimental design. (**B**) Total number of arm entries in Y-maze task. (**C**) Percentage of spontaneous alterations in Y- maze task. **(D**) Escape latency time (s) in Morris water test. Values are mean ± SD, *n* = 8. Ns (no significant) and ^#^
*p* < 0.05, ^###^
*p* < 0.001 as compared with control group; * *p* < 0.05 and *** *p* < 0.001 as compared with scopolamine-treated group by one-way ANOVA.

**Figure 7 molecules-26-02870-f007:**
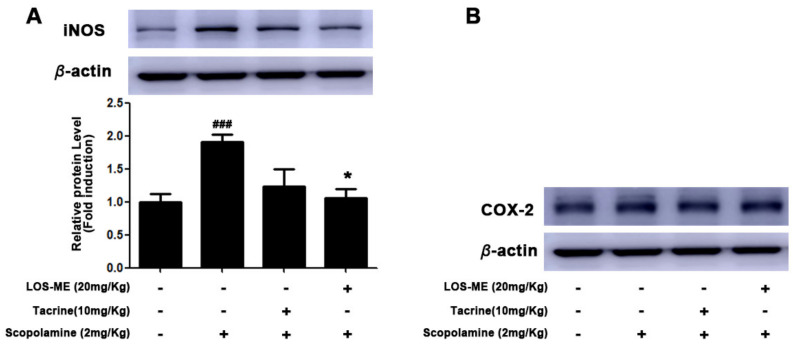
Effect of LOS-ME on iNOS and COX2 expression in scopolamine-induced mouse hippocampus. The hippocampal part of the mouse brain was isolated and confirmed by Western blot. (**A**) iNOS protein expression with corresponding fold change. (**B**) COX2 protein expression. β-actin was used as an internal control. Values are mean ± SD, *n* = 8, ^###^
*p* < 0.001 as compared with control group; * *p* < 0.05 as compared with scopolamine-treated group by one-way ANOVA.

## Data Availability

Data sharing not applicable.

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
