# Peer review of "Mitigating Effect of Lindera obtusiloba Blume Extract on Neuroinflammation in Microglial Cells and Scopolamine-Induced Amnesia in Mice"

_molecules, 2021, doi:10.3390/molecules26102870_

Round 1
Reviewer 1 Report
The authors have studied the effect of Lindere obtusiloba´s methanol extract on neuroinflammation and scopolamine –induced amnesia.
In Asian these kind of traditional herbs are still widely used and well organised scientific studies are certainly needed. However, in most cases the analyses and exact content of these extract have not been characterised. This is the case also in this study. The preparation of the used extract is poorly described; ..shaken time to time…, at which temperature?, solvent was completely evaporated etc. I would expect more precise and more professional preparation and chemical analyses of the extract in order to know the quality of the used extract. Doing this it is possible to compare results with other studies where similar extract has been used. As a very toxic solvent the possible methanol traces should have been analysed from the final extract.
Otherwise the study is well described and performed. However I would expect that the authors describe the aim more clearly. Is the main aim to show that this kind of not-professionally prepared extract as such affect the neuroinflammation and scopolamine induced amnesia? And if this turns out to be true the extract is worth to be studied further? Please clarify the aim.
Comments:
- Please explain the use of Scopolamine and Tacrine; what kind of compounds they are and why they were used in this study. SCOP and TAC abbreviations were mentioned for these compounds. However those were not used in the text itself. Please remove the abbreviations or use throughout the manuscript.
- Please use the mean ± SD instead of mean ± SEM to present all the results. The SD quantifies scatter — how much the values vary from one another. Whereas the SEM describes how precise the mean of the sample is versus the true mean of the population. As the number of animals is low the use of SD is preferable.
- Check also that mL is used instead of ml. Now both notations have been used in the text. The SI-unit symbol for second/seconds is s. Please correct.
- The study timeline is somehow unclear: scopolamine was administered on day 8. Were the Y-maze tests made on the same day? What about the Morris water maze test where the mice were adapted for three days before the actual test? When was this test in relation to scopolamine? Please describe clearly.
- The used amount of LOS-ME is large, 20 mg/kg. Did the authors notice any harmful effect of the use of LOS-ME or was it well tolerated? Were the liver and spleen normal? How about the general well-being of the animals? Please mention this in the text.
- The quality of the figures in the manuscript PDF was not good. They were too small and not sharp. Please resize and check the quality of the figures.
Author Response
Title: Mitigating effect of Lindera obtusiloba Blume extract on neuroinflammation in microglial cells and scopolamine-induced amnesia in mice.
Review 1 Comments:
In Asian these kind of traditional herbs are still widely used and well organized scientific studies are certainly needed. However, in most cases the analyses and exact content of these extract have not been characterized. This is the case also in this study. The preparation of the used extract is poorly described; .shaken time to time…, at which temperature?, solvent was completely evaporated etc. I would expect more precise and more professional preparation and chemical analyses of the extract in order to know the quality of the used extract. Doing this it is possible to compare results with other studies where similar extract has been used. As a very toxic solvent the possible methanol traces should have been analyzed from the final extract. Otherwise the study is well described and performed. However, I would expect that the authors describe the aim more clearly. Is the main aim to show that this kind of not-professionally prepared extract as such affect the neuroinflammation and scopolamine induced amnesia? And if this turns out to be true the extract is worth to be studied further? Please clarify the aim.
Response: We appreciate the interest shown in our article. As suggested a detailed extraction and preparation of the extract is included based on the comments raised in the revised manuscript. Further, we have conducted our study scientifically to investigate the effect L. obtusiloba stem extract in suppressing the neuroinflammation and cognitive impairment in both in vitro and in vivo experimental models.
Comments:
-Please explain the use of Scopolamine and Tacrine; what kind of compounds they are and why they were used in this study. SCOP and TAC abbreviations were mentioned for these compounds. However, those were not used in the text itself. Please remove the abbreviations or use throughout the manuscript
Response: The use of Scopolamine and Tacrine in our study were included in the revised manuscript at appropriate sections (Introduction and discussion section). We deleted the abbreviation.
-Please use the mean ± SD instead of mean ± SEM to present all the results. The SD quantifies scatter — how much the values vary from one another. Whereas the SEM describes how precise the mean of the sample is versus the true mean of the population. As the number of animals is low the use of SD is preferable.
Response: As suggested we use Mean ± SD for all statistical evaluations and included in the study.
-Check also that mL is used instead of ml. Now both notations have been used in the text. The SI-unit symbol for second/seconds is s. Please correct.
Response: As indicated “ml is changed to mL” and we followed uniformity in the entire manuscript.
-The study timeline is somehow unclear: scopolamine was administered on day 8. Were the Y-maze tests made on the same day? What about the Morris water maze test where the mice were adapted for three days before the actual test? When was this test in relation to scopolamine? Please describe clearly.
Response: As suggested, the schedule of animal experiments was included into the figure 6 as Figure 6A.
-The used amount of LOS-ME is large, 20 mg/kg. Did the authors notice any harmful effect of the use of LOS-ME or was it well tolerated? Were the liver and spleen normal? How about the general well-being of the animals? Please mention this in the text.
Response: As indicated we have included the statement regarding the safety of the used extract with giving relevant references in the revised manuscript.
-The quality of the figures in the manuscript PDF was not good. They were too small and not sharp. Please resize and check the quality of the figures.
Response: As indicated the quality of the figures were enhanced to 300 dpi for more clarity.
Reviewer 2 Report
This is a very sound investigation of the potential for Lindera extract to modify a range of biochemical consequences of neuroinflammation. All of the biochemical-physiological studies are well described and the data is entirely convincing.
The work on scopolamine-impaired mice is also sound, but the relationship of this model to the types of cognitive impairment of interest to the authors is questionable. I recognize that there are few good animal models for any of the types of cognitive impairment seen in Alzheimer disease or other age-related neuronal decline syndromes, but the final conclusion (that drugs or extracts of Lindera merit clinical trials) is a bit over the top. Fortunately no reliable drug entity will move immediately in this direction, but it behooves the authors to state clearly that much more animal-based work (including some neuroanatomical and neurochemical study) is necessary at this stage.
Author Response
Title: Mitigating effect of Lindera obtusiloba Blume extract on neuroinflammation in microglial cells and scopolamine-induced amnesia in mice.
This is a very sound investigation of the potential for Lindera extract to modify a range of biochemical consequences of neuroinflammation. All of the biochemical-physiological studies are well described and the data is entirely convincing.
The work on scopolamine-impaired mice is also sound, but the relationship of this model to the types of cognitive impairment of interest to the authors is questionable. I recognize that there are few good animal models for any of the types of cognitive impairment seen in Alzheimer disease or other age-related neuronal decline syndromes, but the final conclusion (that drugs or extracts of Lindera merit clinical trials) is a bit over the top. Fortunately, no reliable drug entity will move immediately in this direction, but it behooves the authors to state clearly that much more animal-based work (including some neuroanatomical and neurochemical study) is necessary at this stage.
Response: We appreciate the interest in our study. As indicated we discussed about the model and also the future studies necessary to confirm the effects. The changes were indicated in red color font for easy identification.